# Fulminant Giant Cell Myocarditis following Heterologous Vaccination of ChAdOx1 nCoV-19 and Pfizer-BioNTech COVID-19

**DOI:** 10.3390/medicina58030449

**Published:** 2022-03-20

**Authors:** Dong-Hoon Kang, Joo-Young Na, Jun-Ho Yang, Seong-Ho Moon, Sung-Hwan Kim, Jae-Jun Jung, Ho-Jeong Cha, Jong-Hwa Ahn, Yong-Whi Park, Sang-Yeong Cho, Ho-Kyung Yu, Soo-Hee Lee, Mi-Yeong Park, Jong-Woo Kim, Joung-Hun Byun

**Affiliations:** 1Department of Thoracic and Cardiovascular Surgery, Gyeongsang National University College of Medicine, Gyeongsang National University Changwon Hospital, 11, Samjeongja-ro, Seongsan-gu, Changwon 51472, Korea; drk82@hanmail.net (D.-H.K.); junhoyah@hanmail.net (J.-H.Y.); hoya_m@naver.com (S.-H.M.); clariboy@naver.com (S.-H.K.); thoracoscope@gmail.com (J.-J.J.); nombrehj@gmail.com (H.-J.C.); cs99kjw@hanmail.net (J.-W.K.); 2Department of Forensic Medicine, Pusan National University School of Medicine, Yangsan 50612, Korea; pdrdream@gmail.com; 3Department of Internal Medicine, Gyeongsang National University College of Medicine and Cardiovascular Center, Gyeongsang National University Changwon Hospital, Changwon 51472, Korea; jonghwaahn@naver.com (J.-H.A.); angio2000@hanmail.net (Y.-W.P.); hiha-ki@hanmail.net (S.-Y.C.); 4Department of Anesthesiology and Pain Medicine, Gyeongsang National University College of Medicine, Gyeongsang National University Changwon Hospital, Changwon 51472, Korea; sciatic@naver.com (H.-K.Y.); lishiuji@naver.com (S.-H.L.); ele93547@naver.com (M.-Y.P.)

**Keywords:** COVID-19, vaccination, giant cell myocarditis

## Abstract

A 48-year-old female patient underwent a heart transplantation for acute fulminant myocarditis, following heterologous vaccination with the ChAdOx1 nCoV-19 and Pfizer-BioNTech COVID-19. She had no history of severe acute respiratory syndrome coronavirus-2 infection. She did not exhibit clinical signs or have laboratory findings of concomitant infection before or after vaccination. Heart transplantation was performed because her heart failed to recover with venoarterial extracorporeal oxygenation support. Organ autopsy revealed giant cell myocarditis, possibly related to the vaccines. Clinicians may have to consider the possibility of the development of giant cell myocarditis, especially in patients with rapidly deteriorating cardiac function and myocarditis symptoms after COVID-19 vaccination.

## 1. Introduction

The global spread of the coronavirus disease 2019 (COVID-19) caused by severe acute respiratory syndrome-coronavirus-2 (SARS-CoV-2) has resulted in countless deaths. To prevent this, several COVID-19 vaccines have been used worldwide. The ChAdOx1 nCoV-19 (or AZD1222) vaccine is an adenoviral vector-based vaccine, and its design was based on the vaccine for the previous Middle East Respiratory Syndrome Coronavirus [1]. Pfizer-BioNTech COVID-19, using mRNA, encodes a pre-fusion membrane-bound stabilized full-length S-2P encapsulated with lipid nanoparticles [2].

Vaccination is crucial in controlling the spread of COVID-19 and decreasing the COVID-19 morbidity and mortality rates. The development of myocarditis and inflammatory myocardial cellular infiltrates have been reported after vaccination, especially after the smallpox vaccine [3].

Witberg et al. reported association between the development of myocarditis and the receipt of messenger RNA (mRNA) vaccines against COVID-19. The estimated incidence of myocarditis was 2.13 cases per 100,000 persons. Most cases of myocarditis were mild or moderate in severity [4]. Two previous studies on myocarditis, following COVID-19 mRNA vaccination, reported cardiac biopsy results, which showed no evidence of myocarditis [5,6]. Almost all patients had resolution of symptoms and signs, and improvement in clinical markers and imaging with or without treatment. In our case, the patient presents a rapid deterioration of cardiac function, following heterologous ChAdOx1 nCoV-19 and Pfizer-BioNTech COVID-19 vaccination and cardiac transplantation. We present a case of fulminant giant cell myocarditis (GCM) following heterologous vaccination with the ChAdOx1 nCoV-19 and Pfizer-BioNTech COVID-19 vaccines and discuss its clinicopathologic findings.

## 2. Case Report

A 48-year-old woman visited the emergency department due to dull chest pain and dyspnea. She had received the initial dose of ChAdOx1 nCoV-19 vaccine 77 days before she was vaccinated with a second dose of BNT162b2 mRNA COVID-19 vaccine. Four days after the second dose, she felt the symptoms. She had a medical history of hypothyroidism, and she had a negative family history of rheumatologic or genetic diseases.

On physical examination, she had no fever and normal blood pressure (104/60 mmHg), but she had tachycardia (122 bpm). Laboratory testing during the first 24 h revealed an N-terminal pro b-type natriuretic peptide level of 6631 pg/mL (0.0–125), peak highly sensitive troponin-T of 6820 ng/L (0–14), creatine kinase (CK)-MB of 66.5 ng/mL (0.0–3.6), CK of 415 U/L (0–145), lactic acid of 3.4 mmol/L (0.5–2.2), and C-reactive protein of 9.6 mg/L (0–5). Table 1 illustrates the extensive infectious workup performed in this case. The patient had no evidence of prior bacterial or viral infection.

The initial chest radiography had unremarkable findings (Figure 1A). Initial electrocardiogram showed marked ST segment elevation in V2–V5 precordial lead and sinus tachycardia (Figure 1C). Transthoracic echocardiography revealed global akinesia with severe left ventricular (LV) dysfunction (ejection fraction 15%) and right ventricular dysfunction. Coronary angiography (CAG) showed severe stenosis (~90%) at the proximal to left anterior descending artery (LAD) (Figure 1D). The right coronary artery had no disease. Despite the good coronary flow (thrombolysis in myocardial infarction, flow grade 3), we implanted a drug-eluting stent at the proximal to mid LAD because the ST-segment elevation myocardial infarction was not completely excluded. During percutaneous coronary intervention, sudden ventricular tachycardia occurred, and extracorporeal cardiopulmonary resuscitation was performed (Figure 1B).

Cardiac magnetic resonance imaging was not performed because of the patient’s poor condition. Her heart did not recover, and heart transplantation was performed 11 days later. Organ autopsy of the explanted heart revealed giant cell myocarditis (GCM). The patient did not experienced a rejection or recurrence for 8 months after heart transplantation. The patient was discharged without complication and is currently following up at an outpatient clinic.

## 3. Discussion

The development of symptoms four days after the patient received her second vaccine dose, severe heart failure inconsistent with CAG findings, and negative viral serum markers highly suggested a relationship between the patient’s heart condition and vaccination. Specifically, the clinical picture of our patient was deteriorating too rapidly.

Shay DK. et al. report that similar clinical characteristics of the patients, recent COVID-19 vaccination, and insufficient evidence to support other acute myocarditis etiologies suggest an association with immunization [7]. In addition, the development of acute-onset chest pain three to five days after vaccination, particularly the second dose, was a typical feature of an immune-mediated reaction [2]. Although the mechanisms underlying the development of myocarditis following vaccination are unclear, molecular mimicry between the S protein of SARS-CoV-2 and self-antigens pre-existing dysregulated immune pathways triggered in certain individuals’ immune response to mRNA and activation of immunological pathways, and dysregulated cytokine expression have been proposed [8].

Previous cardiac histopathologic studies have reported the absence of diffuse lymphocytic myocarditis traditionally seen in viral myocarditis and the presence of CD68+ macrophage identified in the myocardium, and they were ultrastructurally characterized by cytopathy, with membrane damage and cytoplasmic vacuoles [9]. Ten hearts of patient who died from COVID-19, including five myocarditis cases, revealed that there was no evidence of lymphocytic myocarditis. Additionally, they showed a greater number and diffuse distribution of CD68+ cells compared with matched control or other myocarditis hearts, indicating that cells of monocyte/macrophage lineage rather than lymphocytes may be dominant in this setting [10]. Furthermore, SARS-CoV-2 viral particles have also been detected in interstitial cytopathic macrophages in the myocardium, which suggests that SARS-CoV-2 can reach the heart during viremia or through the infiltration of infected macrophages into the myocardium [9]. These reports suggest that macrophages in the myocardium could play a significant role of development of myocarditis in the COVID-19.

The patient’s heart did not recover, and heart transplantation was performed. Pathologic examination of the explanted heart revealed diffuse cardiomyocyte necrosis and mixed inflammation in the atria, ventricles, and interventricular septum. The mixed inflammatory infiltrations consisted of lymphocytes, macrophages, and eosinophils. Furthermore, scattered multinucleated giant cells, which were immunohistochemically reactive for CD68, were observed. Granulomatous lesions were not observed. These findings are consistent with GCM (Figure 2). Inflammatory infiltration was also noted in the sinoatrial and atrioventricular nodes. The immunohistochemical examination was negative for cytomegalovirus infection.

The etiology of GCM is uncertain to date. However, animal model and human studies have provided evidence that GCM is an autoimmune disorder, mediated largely by T-lymphocyte activity associated with macrophage antigens from giant cells [11]. Diffuse cardiomyocyte necrosis and mixed inflammation in the present case, including multinucleated giant cell, suggest a maladaptive immune-mediated myocardial injury. This diffuse necrosis of cardiomyocytes may be caused by both ACE2-mediated and molecular mimicry between the S protein of SARS-CoV-2 and self-antigens. Some autoantibodies, such as antinuclear antibody, anticardiolipin antibody, and lupus anticoagulant were detected in patient with COVID-19. Because SARS-CoV-2 can break immune tolerance and induced autoimmune response, it is also likely to induce clinical autoimmunity [12]. Furthermore, in certain individuals with genetic predisposition, the immune response to mRNA may not be turned down and may drive the activation of an aberrant innate and acquired immune response [13]. We suggest that this innate immunity and adaptive immune response probably could provoke GCM in genetically susceptible people who administrated heterologous adenoviral vector-based and mRNA vaccination.

Bobbio et al. showed that heart transplantation has been the only definitive treatment for advanced GCM, and there was no difference in 1-year survival rates after heart transplantation with and without GCM [14]. In addition, Kandolin et al. showed that repetitive endomyocardial biopsies are required for the diagnosis of GCM, and partial clinical remission is shown in 2/3 of patients with immunosuppression [15].

The present case highlights an important clinical point. The patient presented with a rapidly deteriorating cardiac function, following heterologous vaccination with ChAdOx1 nCoV-19 and Pfizer-BioNTech COVID-19 vaccines. Based on the pathological examination of her heart, she was diagnosed with GCM.

## 4. Conclusions

Although the pathogenesis of GCM, following heterologous vaccination, is unclear, clinicians should be aware of the possibility of GCM development especially in patients with rapidly deteriorating cardiac function and myocarditis symptoms after COVID-19 vaccination. Despite this rare potential vaccine-related adverse effect, the benefits of COVID-19 vaccination outweigh the risks.

## Figures and Tables

**Figure 1 medicina-58-00449-f001:**
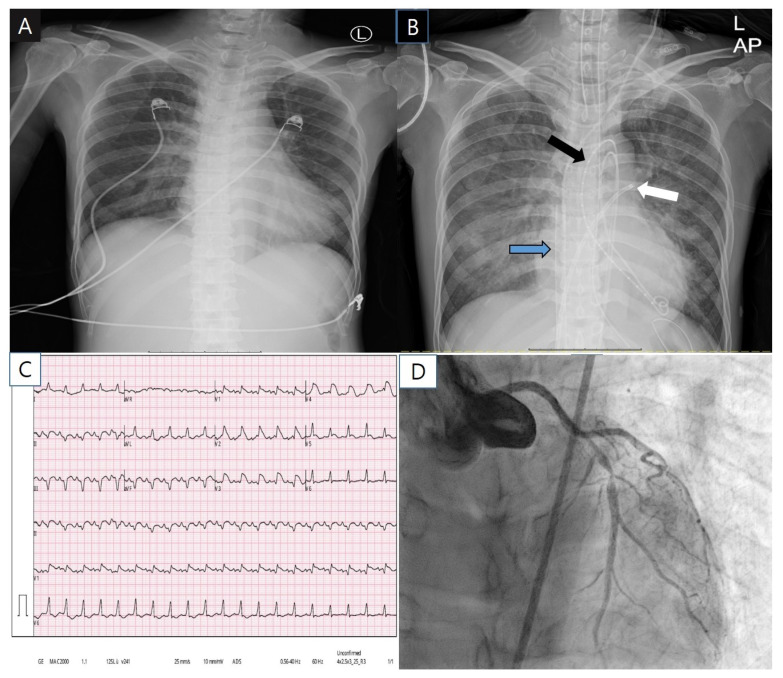
The patient’s chest X-ray (**A**,**B**), electrocardiogram (**C**) and coronary angiography (**D**). Black arrow, trans-aortic LV venting catheters; white arrow, transseptal left atrium venting catheter; Blue arrow, venous drainage catheter of extracorporeal oxygenation support.

**Figure 2 medicina-58-00449-f002:**
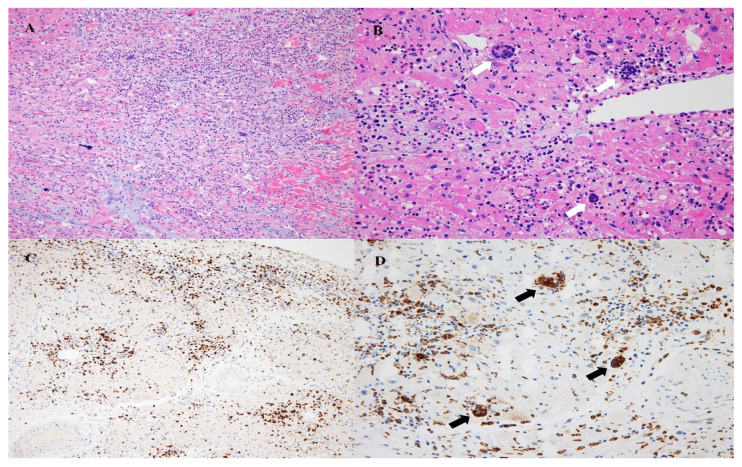
Pathologic findings of the heart. Diffuse cardiomyocyte necrosis and mixed inflammatory infiltration are noted (**A**; H&E, ×100). Mixed inflammation including lymphocytes, macrophages, and frequent eosinophils are noted, and multinucleated giant cells are also noted (**B**; H&E, ×200). T-lymphocytic infiltration is identified (**C**; CD3, ×100). Infiltration of CD68+ macrophages is identified, and multinucleated giant cells are positive for CD68 (**D**; CD68, ×200). Black and white arrows show multinucleated giant cells.

**Table 1 medicina-58-00449-t001:** Viral and bacterial studies of the patient.

Viral Study	
Respiratory Syncytial virus	Negative
Parainfluenza 1,2,3,4	All negatives
Influenza A	Negative
Influenza B, A-H3, A-H1-pan, A-H1-2009	All negatives
Human Rhinovirus	Negative
Human Enterovirus	Negative
Human Metapneumovirus	Negative
Adenovirus	Negative
Coronavirus OC43, NL63, HKU1, 229E	All negatives
HAV, HBV, HCV	All negatives
HIV	Negative
RPR	Negative
SARS-CoV-2	Negative
Bacterial study	
Blood culture	Negative
Mycoplasma pneumonia	Negative
Chlamydophila pneumonia	Negative
Bordetella pertussis	Negative

HAV, hepatitis A virus; HBV, hepatitis B virus; HCV, hepatitis C virus; HIV, human immunodeficiency virus; RPR, rapid plasma reagin test; SARS-CoV-2, Severe acute respiratory syndrome coronavirus 2.

## Data Availability

The study did not report any data.

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
