# Peer review of "Fulminant Giant Cell Myocarditis following Heterologous Vaccination of ChAdOx1 nCoV-19 and Pfizer-BioNTech COVID-19"

_medicina, 2022, doi:10.3390/medicina58030449_

Round 1

Reviewer 1 Report

There is no doubt that it is a quite intriguing case report, suggesting the possible implication of anti-COVID-19 vaccination in the pathogenesis of giant cell myocarditis (GCM). Actually, despite a critical concomitant coronaropathy, the pathologic examination on explanted heart provided clear histological and histochemical evidence that the patient had a giant cell myocarditis. Nevertheless, the search for genome of cardiotropic viruses in myocardial tissue wasn’t carried out. It would be interesting if the authors could provide also this information. However, due to its rarity, the association between anti-COVID-19 vaccination and the development of GCM probably remains anecdotal.

Reviewer 2 Report

This case report does not add anything new or innovative to what is already scientifically known.

Carrying out an autopsy on a patient's heart can be informative if the hypothesis and confirmation are in agreement. Unfortunately, the conclusions are also uncertain and unclear.

The authors did not find sufficient reasons to support the vaccine causing myocarditis. The etiology of this pathology is currently unknown and making such assumptions could be a gamble.

Reviewer 3 Report

The paper is interesting but, should be able to answer the a few queries:

  • I understand English may not be the first language of the authors, but to ensure higher quality publication matching the journal standards native English speaker should go through the manuscript.
  • I suggest to shorten the introduction (lines 36-45) since too many details about vaccines are included.
  • Although GCM has often a fulminant course, a treatment strategy has been developed. I encourage to include some details about diagnostic work-up and treatment of GCM
  • Line 59: the word "fatal" seems inappropriate considering that the patient is still alive. Maybe "fulminant" could be a better choice
  • Lines 63-68:  the authors are encouraged to be more specific about the character of the pain. Furthermore, the repetition of "chest pain and dyspnea" in line 63 and 65 is redundant.
  • Lines 69-70: please reformulate
  • did the patient need ECM or any other MCS?
  • line 97: It would be interesting to have more details about the current status of the patient. How many days/months have passed since the heart transplantation (HT)? there is still uncertainty about the outcome after HT because of GCM. Have the patient experienced any rejection or recurrence? 
  • lines 137-140: please reformulate. GCM is not limited to middle-aged female. Still I suggest to not draw strong conclusions based on a single case.
  • It has been recently shown that post-HT outcome in patients with GCM is not worse than other etiologies (Bobbio et al. 2022 Feb;111(2):125-140). This, together with a few details about treatments, needs to be named in the discussion section.  

Round 2

Reviewer 2 Report

The article is no different from the previous version. This case report does not provide any information, it is just a description of what happened to a patient. Is it related to Covid19? Could the author confirm and how could they affirm that there is a relationship between symptoms, intervention, therapy and Covid? They do not explain how it is possible to correlate everything and what are the molecular mechanisms underlying the whole story. 
